# Patient Specific Numerical Modeling for Renal Blood Monitoring Using Electrical Bio-Impedance

**DOI:** 10.3390/s22020606

**Published:** 2022-01-13

**Authors:** Mugeb Al-harosh, Egor Chernikov, Sergey Shchukin

**Affiliations:** Department of Medical and Technical Information Technology, Bauman Moscow State Technical University, 105005 Moscow, Russia; ches16l294@student.bmstu.ru (E.C.); schookin@bmstu.ru (S.S.)

**Keywords:** renal blood circulation, electrode system, numerical modeling, electrical impedance

## Abstract

Knowledge of renal blood circulation is considered as an important physiological value, particularly for fast detection of acute allograft rejection as well as the management of critically ill patients with acute renal failure. The electrical impedance signal obtained from kidney with an appropriate electrode system and optimal electrode system position regarding to the kidney projection on skin surface reflects the nature of renal blood circulation and tone of renal blood vessels. This paper proposes a specific numerical modelling based on prior information from MRI-data. The numerical modelling was conducted for electrical impedance change estimation due to renal blood distribution. The proposed model takes into the account the geometrical and electrophysiological parameters of tissues around the kidney as well as the actual blood distribution within the kidney. The numerical modelling had shown that it is possible to register the electrical impedance signal caused by renal blood circulation with an electrode system commensurate with the size of kidney, which makes it possible to reduce the influence of surrounding tissues and organs. Experimental studies were obtained to prove the numerical modelling and the effectiveness of developed electrode systems based on the obtained simulation results. The obtained electrical impedance signal with the appropriate electrode system shows very good agreement with the renal blood change estimated using Doppler ultrasound. For the measured electrical impedance signal, it is possible to obtain the amplitude-time parameters, which reflect the hemodynamic characteristics of the kidneys and used in diagnostics, which is the subject of further research.

## 1. Introduction

Renal function is intimately dependent on renal blood circulation [1]. However, the measurement of renal blood circulation is time consuming and not widely available, thus the current methods are either invasive, rely on the use of radioactive tracers which are diffusible, have kinetics which are complicated by tubular transport or glomerular filtration, or cannot allow the assessment of small variations and rapid changes in renal blood flow and cannot be used for bedside monitoring [2]. The real time measurement of renal blood flow using contrast-enhanced ultrasound was considered in [2,3], although, this method is cost-effective and widely available, it still has considerable limitations in sensitivity and specificity for the diagnosis of allograft rejection [3]. Renal near-infrared spectroscopy is used to evaluate regional oximetry in a non-invasive continuous real-time fashion, and reflects tissue perfusion [4], however the near-infrared method has a depth limitation, which makes this approach applicable only to special patients like newborns.

Several studies on animals and subjects was conducted to prove that the electrical impedance signal obtained from kidney correlate with the change of blood circulation and reflect the tone of blood vessels [5,6]. Nevertheless, due to the anatomical location of kidney especially with obese patients as well as the adjacent organs such as liver, spleen even the surrounds muscles and fat tissues made the authors, who studied this method previously to conclude that the electrical impedance signal obtained using skin attached electrodes gives a characteristic of the complex blood flow change of all structures located in the inter-electrode space. For precise assessment, implantable electrical impedance device, which can provide measurements on wide range of frequencies was considered to diagnose the rejection in kidney transplantation in previous studies [5,7].

However, in order to achieve high accuracy of determining the amount of blood per 100 G of kidney tissues using attached non-invasive electrodes, it is required to develop a detailed specific mathematical model of the region of interest as well as a method to eliminate the effect of surrounding organs and tissues. Improvement of this technique can allow its application for monitoring patients undergo different type of cardiovascular surgeries, who need a monitoring to prevent the acute kidney injury, which is a common complication after cardiac surgery and is associated with increased morbidity and mortality and increased length of stay in the intensive care unit [8]. Furthermore, the develop of such system with precise measurements can be used for acute kidney allograft rejection diagnosis. In related previous work [9] belongs to the authors of this paper, different biophysical models were considered to describe the kidney and take in to the account the electro physiological parameters, and the depth of kidney. Nevertheless, these models are more reliable for non-obese patients and children because as the kidneys are located deeper the impact of the surrounding tissues become greater and hence leads to an error of determining the amount of blood per 100 G of kidney tissues. However, the desirable area to acquire the electrical impedance signal can be presented as a multi-layered model; these layers specify the biological tissues of the region of interest, which are: the skin, subcutaneous fat layer, the soft muscle layer, the fat layer around the kidney, and the kidney layers. According to the analysis of the electrical properties of biological tissues at several excitations frequencies that has been considered in related works [9,10], it was concluded that the most adequate frequency range is 80–100 kHz where active resistance is the main component of impedance. Thus, at lower frequencies the impact of capacitive part increases, which is determined by the heterogeneity of tissue structures in the region of interest, furthermore the low frequencies can cause seizures and polarizing electronic phenomena. Thus, an electrical current with 100 kHz has been used during this study. Figure 1 shows the electrical resistivity of biological tissues in the study area in different frequencies including 100 kHz [11]. Four electrode system configuration was proposed in this study to provide the desired accuracy for bio-impedance measurement, thus two outer electrodes are used to inject a small alternating current into the human body the voltage drop is measured to calculate the complex impedance [12].

However, the aim of this study is to conduct a numerical modeling based on prior information obtained from MRI-images. This modelling should take into the account the geometrical and electrophysiological parameters of the tissues above the kidney as well as the actual kidney dimensions and blood distribution in order to estimate the contribution of the surrounding tissues around the kidney located in the inter-electrode space. Additionally, the task of verifying the developed model with the help of an experimental study was set.

## 2. Materials and Methods

Numerical modeling has been conducted using a simulation platform COMSOL Multiphysics 5 [13,14,15]. MRI images have been used as an input for 3-D model creation. The preliminary processing was carried out using MicroDicom [16]. In order to reduce the calculation time, the right kidney was considered during the numerical modelling. During this study, the actual kidney size, the dimensions of the patient’s torso, the thickness of subcutaneous fat and muscle layers, as well as the distance of the kidney relative to the center of the trunk cross-section were outlined as shown in Figure 2.

The 3-D model was implemented in COMSOL Multiphysics, while the electrical resistivity of tissues in the region of interest was set according to Gabriel et al. [11]. Figure 3 shows the obtained 3-D model of perineal space with an electrode system located at the center of right kidney.

In this model, the kidney is represented by a two-layer ellipse. These layers mimic the renal cortex and medulla. The kidney is located in the fibrous layer, presented in the form of a cylinder with major and minor axes (2a_1_ − 2h_m_) and (2b_1_ − 2h_m_), respectively. The electrodes are modeled as cylinders equidistant from each other with a base diameter of 3 mm. Table 1 shows the geometrical dimensions of the designed model obtained from MRI-images.

The change of kidney electrical resistivity due to the blood filling during cardiac cycle has been estimated using the parallel conductor model, illustrated in Figure 4 [17].

As the kidneys receive a total blood flow of approximately 1000 mL per minute (20% of the cardiac output). This equates to 300–400 mL per minute per 100 g of tissue [18]. The Equation (1), which deduced from the parallel conductor model, has been used to calculate the electrical resistivity change of healthy kidney which receives a total blood flow of approximately 400 mL per minute per 100 g of tissue.
(1)Δρ=ρk − ρtotal=(ρk ∗ mb (ρk − ρb))/(mk ∗ ρb + mb ∗ ρk)
where ρk is the electrical resistivity of kidney tissue, ρb is the electrical resistivity of blood; ρtotal is the electrical resistivity of a blood-filled kidney.

In previous work [10] it has been concluded that the estimation using the parallel conductor model showed that the dynamic electrical resistivity change of the kidney due to nature of renal blood circulation in normal cases is approximately 1 Ω·m and the cortical as well as the medulla layers perceive such a change in different percentages according to the data illustrated in Table 2. The table shows the static electrical resistivity of kidney layers before and after blood filling, two different cases were considered in this study to estimate the sensitivity of electrical impedance measurement to different blood flow distribution in kidney layers. The first case: the cortex receives a 90% of the blood flow to the kidney which corresponds to normal perfusion while the second case suggests less perfusion of cortex.

During the numerical simulation, in line four electrodes with equal distance between the adjacent electrodes was used, thus the distance between the measuring electrodes (b) was varied in the range from 7 to 40 mm with a step of 3 mm. Accordingly, the distance between the current electrodes, which is 3 × b has been changed. The static electrical impedance values as well as the bio-impedance change have been calculated for every electrode system iteration change. The different cases of layered distribution of renal blood flow were considered during the modeling.

The experimental studies on subject were carried out in the medical and technological center of Bauman Moscow state technical university. The multichannel system (REO-32) has been used for electrical impedance measurements. The system provides 4-electrode technique by applying alternating current of frequency 100 kHz and constant amplitude to a current electrodes 3 mA [19]. The technical specifications of REO32-system are shown in Table 3.

The experimental studies were carried out on the same subject, whose MRI images were used for numerical modeling. In the study, 3 different electrode systems were designed as shown in Figure 5. The distances between the current (2a) and measuring (2b) electrodes for each electrode system are illustrated in Table 4.

Figure 6 shows the electrode system positioning during the experiment. The studies were carried out in the area of the right kidney and electrical impedance mapping were performed with 20 mm shifts to the right due to the design dimensions of the electrode system used. In order to reduce the influence of kidney displacement during breathing, the measurements were taken with breath holding. The measurements were carried out several times to confirm the reproducibility of the obtained signals; uniform conditions as well as the degree of pressing of the electrode system were provided.

## 3. Results

### 3.1. Simulation and Experimental Results

The presented plots in Figure 7 show the results of numerical modeling of two cases of layered renal blood distribution. In Figure 7a we show the static electrical impedance change in relation to the electrode spacing change, thus the impedance change is a function which depends on the apparent resistivity of tissues in the reign of interest as well as the geometrical factor of the electrode system. In the case of using a symmetric tetrapolar measurement system, the impedance between the measuring electrodes can be represented with Equation (2) [20]:(2)$$Z\;=\frac{2\rho b}{\pi \left(a^{2}-b^{2}\right)}$$

The significant contribution on static electrical bio-impedance in the area of interest is the fat layer as it has the largest electrical resistivity. The involved subject in this study is 35 year man and weight 55 kg, thus as the electrode spacing increases the impedance decreases for every iteration change which mean that more conductive layers are captured. The corresponding plots of the dynamic electrical impedance change due to blood filling is shown in Figure 7b, during this study it was suggested the origin of the recorded electrical impedance change is the renal blood circulation, thus the influence of the layers above the kidney was neglected during the numerical modelling. In contract with static impedance, the dynamic electrical impedance increases due the increase of the electrode spacing, this indicates the increase of sensitivity to kidney blood filling.

The electrical impedance change for the considered two cases in this study is illustrated in Table 5 with details. The presented values show a high sensitivity of electrode systems with distances of 20 mm and more between the measuring electrodes to the changes in electrical impedance of kidney due to its blood filling, since the probing depth also increases with an increase in the inter-electrode distance [21]. The results analysis shows an impedance value in a range of 140 mΩ can be obtained, which enough for further inverse solution to estimate the hemodynamic parameters of kidney.

The big issue of renal blood flow estimation is the kidney displacement during respiration, thus this displacement can reach few centimeter depending on the breathing depth [22]. The kidney displacement in one direction was considered during this study and modelled by the increase of kidney depth, which can be occurred during inhale. Figure 7c shows the change of electrical impedance due to different kidney depth, hence for real time monitoring this influence should be considered. However, in normal cases when the kidney is located far from the body surface the small electrode system cannot provide the desire sensitivity to kidney blood circulation as shown in Figure 7d, thus the desired impedance signal can be achieved just with increasing the electrode spacing, this increase leads to an increase of the surrounding tissues, however the multi-channel measurement with different electrode spacing can be a solution to increase the accuracy of determining the amount of blood per 100 G of kidney tissues.

The result of electrical bio-impedance recording is shown in Figure 8, before measurement the location of kidney was identified by ultrasound system, the skin was preliminarily treated with isopropyl alcohol (2-propanol, isopropanolm, IPA) 70% (*w*/*v*) to remove the stratum corneum and increase the conductivity. The same electrode system location was achieved as proposed on numerical modeling, the subject was asked to hold breathing for 10 s which is enough to obtain a series of cardiac cycles to make sure of signal reliability. The first measurement was obtained using the smallest electrode system; the static electrical impedance as well as the dynamic was acquired. The same measurement was repeated with the second and the third electrode systems. As shown from the obtained data, the signal becomes clear and more specific to renal blood circulation with appropriate electrode spacing selection, which confirms the requirement of patient specific modelling before measurements to select the appropriate electrode system as well as the right position for every patient individually. The results of experimental studies on subject and illustrated in Table 6 shows the same behavior of numerical modelling, thus the amplitude of pulsatile impedance becomes greater and specifically clear with electrode spacing changes, however the observed difference between the experimental and the modelling values is because of the electrical resistivity of biological tissues used during the numerical simulation differ from the real values as the first were taken from data obtained in vitro. However, this difference should be adjusted during the further inverse problem solution.

### 3.2. Verification with Doppler Ultrasound

To assess the contribution of pulse changes in the blood filling of the kidney vessels to the electrical impedance signals, Doppler ultrasound of the kidney vessels was performed. The General Electric models LOGIC S8 with an ML6-15-D ultrasonic transducer at a frequency of 6.3 MHz. was used during this study to obtain prior information such as renal artery and renal vein velocity as well as the corresponding diameters. Breath-holding was employed during ultrasound acquisitions. Figure 9 shows the recoded curves of blood velocities (cm/s) in the renal artery V_a_ and renal vein V_v,_ the corresponding cross sectional area of the renal artery and renal vein are S_a_ = 0.30 cm^2^ S_v_ = 0.38 cm^2^ respectively. However, the kidney blood volume change per one cardio cycle Q(t) (cm^3^) can be calculated based on the difference in the volumetric inflow and outflow according to the Equation (3).
(3)$$Q\left(t\right)=\int_{0}^{t}V_{a}\left(t\right)*S_{a}\mathrm{dt}-\int_{0}^{t}V_{v}\left(t\right)*S_{v}\mathrm{dt},$$

Using the obtained curves, it is easy to achieve to the dependences of the volumetric flow rate. Thus, the values of the blood flow velocities at each time point were multiplied by the cross sectional area of the renal artery and renal vein to get the corresponding volumetric flow rate for renal artery and renal vein as shown Figure 10. The obtained volumetric blood flow curves were used for the integral calculation of the change in blood volume over one cardio-cycle.

Figure 11 show the result of the bio-impedance pulse behavior comparing with the blood volume change within the kidney obtained using ultrasound.

The correlation coefficient between the change in kidney blood volume, calculated from the results of ultrasound measurements, and the change in the electrical impedance during one cardio cycle is 0.89. This results show that the electrical impedance signal obtained with attached electrode reflects the nature of blood circulation in the kidney and showed high correlation with the renal blood behavior of kidney.

## 4. Discussion

Electrical Impedance is a simple and non-invasive method to acquire data on hemodynic parameter of kidney. However, the accuracy of determining this parameter can be affected by various dynamic sources. The origin of these sources can be the adjacent organs and tissues around the kidney as well as the kidney displacement during respiration. Thus, the analysis of the geometrical and electrophysiological parameters of the kidney and tissues around shows that for the study of signal mechanisms formation and the transition from measurable electrical quantities to hemodynamic parameters it is required to develop a justified mathematical model which characterizes the region of interest with full details. The simple analytical models do not fully describe the area of study, which leads to a significant error when solving the inverse function, thus the proposed model in this study takes into account the geometric and electrophysiological parameters of the tissues located around the kidney, as well as the actual distribution of blood between the layers of the kidney. The goal of this study was to develop a patient specific model of perirenal space based on MRI data, which can be used to investigate the mechanisms of electrical bio-impedance signal obtained from kidney using skin attached electrodes. Numerical modeling has shown that it is possible to record the electrical impedance signal caused by renal blood circulation using a specially selected electrode system, commensurate with the size of the kidney and located relative to the longitudinal axis of the kidney, which makes it possible to reduce the influence of surrounding tissues and organs. The kidneys can move during respiration in a tilted coronal and sagittal plane [22], however in order to reduce the influence of kidney replacement during breathing, the measurements were taken with breath holding.

The experimental study on subject with different electrode system shows that electrical impedance signal quality can be influenced by the electrode spacing and some specific points can be disappear in case of measurement with no appropriate electrode system, thus previously identified electrode spacing based on the forward problem solution obtained by the numerical problem can avoid this problem. An important point in this work is the correlation between the registered electrical impedance signals with the arising pulse blood flow fluctuations from kidney obtained using Doppler ultrasound.

## 5. Conclusions

The paper proposed a new approach which allows the monitoring of renal blood circulation based on electrical impedance measurement from specific location regarding to the kidney. Specific numerical modeling has been conducted in order to choose the appropriate electrode system parameters, thus the electrical impedance signal which reflects the nature of renal blood circulation for every patient depends on such parameters. Taking in to the consideration the impact of the surrounding tissues by developing such specific model allows increasing the accuracy of measurement and the hemodynamic parameters, which is the subject of further study.

## Figures and Tables

**Figure 1 sensors-22-00606-f001:**
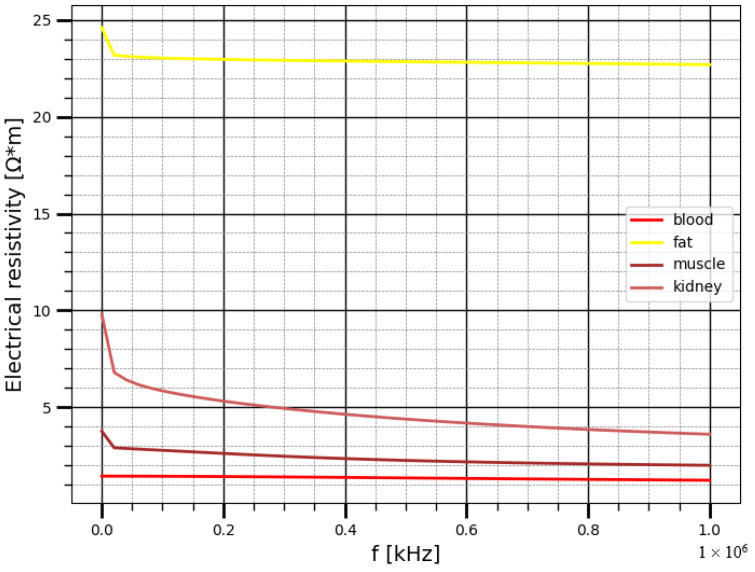
The electrical resistivity of the biological tissues in the study area.

**Figure 2 sensors-22-00606-f002:**
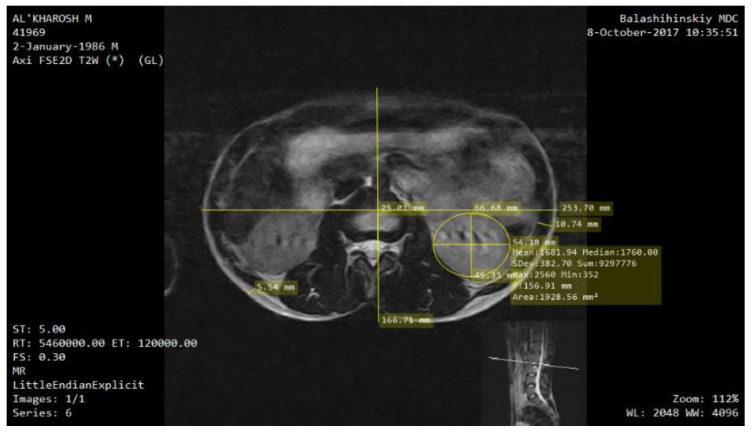
Assessment of the kidney location, its geometric dimensions and the thickness of the perirenal layers according to the MRI image.

**Figure 3 sensors-22-00606-f003:**
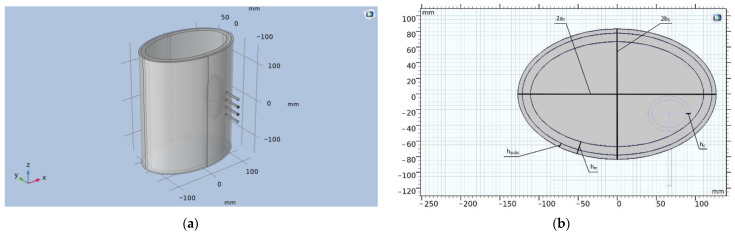
(**a**) Model of the perirenal space in the COMSOL Multiphysics 5.5 software; (**b**) Geometric parameters of the model (cross section), where 2a1—is the major axis of the ellipse (cross section of the trunk), 2b1—is the minor axis of the ellipse (cross section of the trunk), hsubc—is the thickness of the subcutaneous fat layer, hm—thickness of the muscle layer, hc—thickness of the renal cortex.

**Figure 4 sensors-22-00606-f004:**
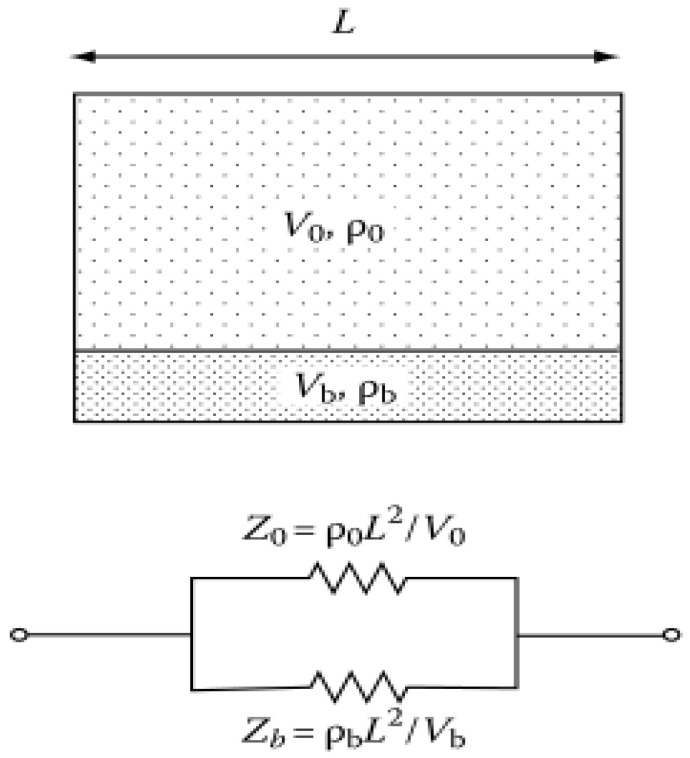
The parallel conductor model and the equivalent circuit, Where: V0, Vb The volume of kidney and blood respectively, ρ0, ρb—the electrical resistivity of kidney tissue and blood respectively.

**Figure 5 sensors-22-00606-f005:**
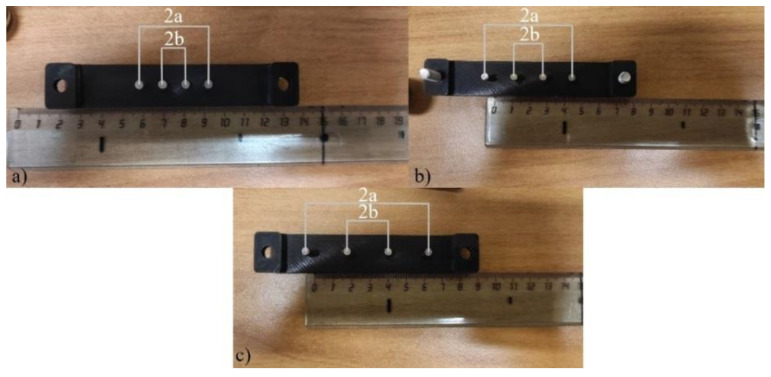
The designed electrode systems: (**a**) the first, (**b**) the second, (**c**) the third.

**Figure 6 sensors-22-00606-f006:**
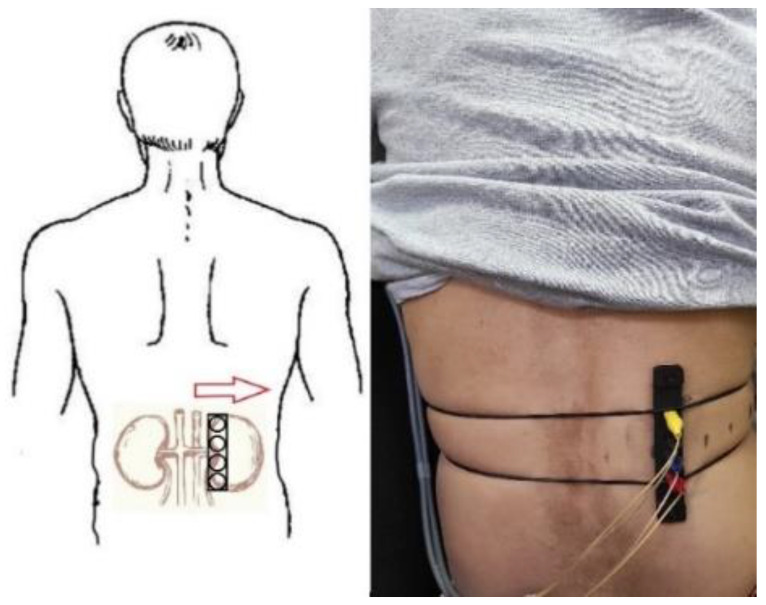
The electrode system position corresponding to the projection of kidney on skin surface.

**Figure 7 sensors-22-00606-f007:**
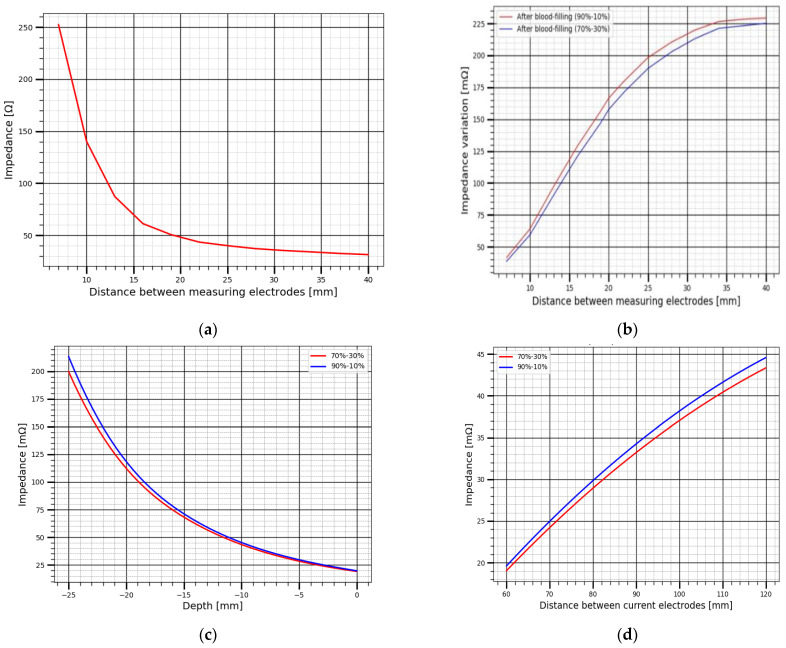
The change of electrical impedance obtained by numerical modelling: (**a**) the variation of static impedance with increasing distance between measuring electrodes, (**b**) the variation of pulsatile impedance with increasing distance between measuring electrodes, (**c**) the variation of pulsatile impedance with increasing kidney depth, (**d**) the variation of pulsatile impedance with increasing distance between measuring electrodes at the maximum kidney depth.

**Figure 8 sensors-22-00606-f008:**
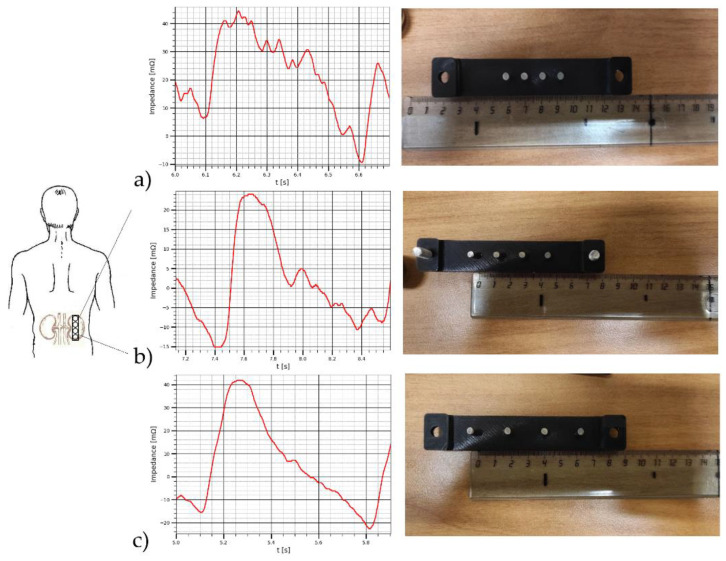
The electrical impedance signals obtained by (**a**) the first electrode system, (**b**) the second electrode system, (**c**) the third electrode system.

**Figure 9 sensors-22-00606-f009:**
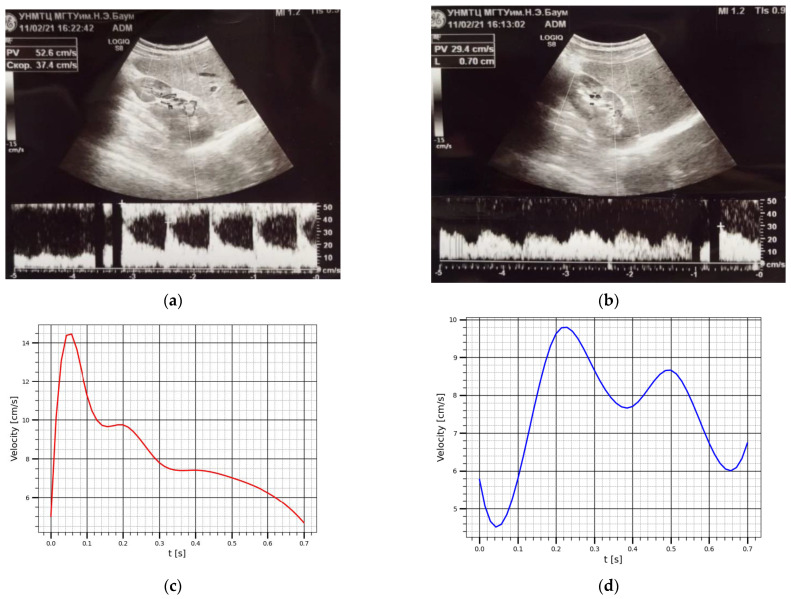
This velocity curves obtained using ultrasound: (**a**) The renal artery blood velocity; (**b**) The renal vein blood velocity; (**c**) The averaged renal artery blood velocity; (**d**) The averaged renal vein blood velocity.

**Figure 10 sensors-22-00606-f010:**
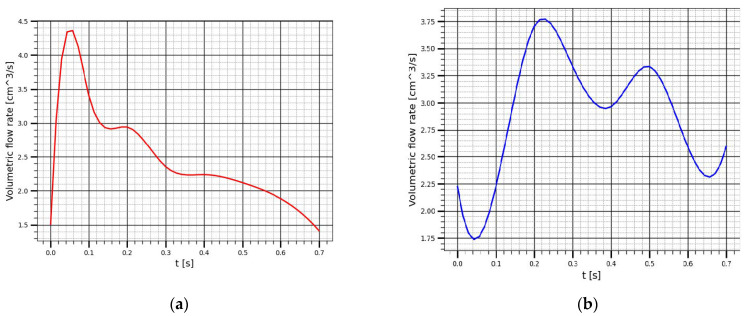
This volumetric flow rate in: (**a**) The renal artery; (**b**) The renal vein.

**Figure 11 sensors-22-00606-f011:**
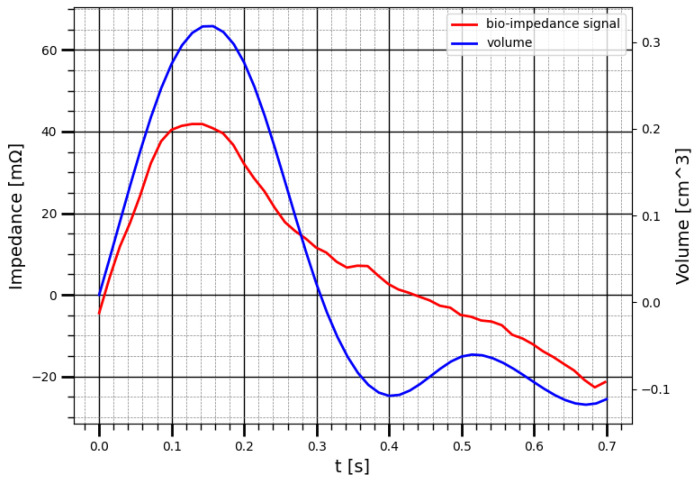
The correlation between bio-impedance signal and the blood volume change of kidney using ultrasound.

**Table 1 sensors-22-00606-t001:** The geometrical parameters of the designed model.

Model Parameters	Value [mm]
2a_1_	253.7
2b_1_	166.71
h_subc_	5.54
h_m_	10.74
h_c_	5

**Table 2 sensors-22-00606-t002:** The electrical resistivity of kidney tissues a according to blood distribution.

Cases	Electrical Resistivity ρ [Ω·m]
Before blood-filling (cortex)	5.84
Before blood-filling (medulla)	5.84
After blood-filling (90–10%, cortex)	4.99
After blood-filling (90–10%, medulla)	5.75
After blood-filling (70–30%, cortex)	5.18
After blood-filling (70–30%, medulla)	5.56

**Table 3 sensors-22-00606-t003:** The technical specification of the bio-impedance system.

Specification	Range
Channels number	32
the current	3 mA
the sampling frequency	500 Hz
frequency band	0.01–137 Hz
the measuring range value	1–250 Ω
ADC	12 bit
Calibration	integrated
The accuracy of pulsatile impedance	1.0 mΩ
The accuracy of static impedance	50 mΩ

**Table 4 sensors-22-00606-t004:** The electrode system parameters.

The Electrode System	2a [mm]	2b [mm]
1st	30	10
2nd	45	15
3rd	60	20

**Table 5 sensors-22-00606-t005:** The obtained electrical impedance values during the numerical modeling.

Distance between Measuring Electrodes [mm]	Before [Ω]	After (90–10%) [Ω]	After (70–30%) [Ω]
7	394.707	394.665	394.669
10	199.349	199.284	199.289
13	111.898	111.801	111.808
16	70.642	70.513	70.521
19	55.924	55.768	55.777
22	46.879	46.699	52.492
25	43.224	43.026	43.034
28	40.481	40.271	40.278
31	38.891	38.671	38.678
34	37.802	37.576	37.581
37	36.589	36.361	36.366
40	35.729	35.499	35.503

**Table 6 sensors-22-00606-t006:** The results of bio-impedance measurement on subject.

Distance between the Current Electrodes [mm]	Base Impedance [Ω]	dZ [mΩ]
30	95.002	32.94
45	69.448	43.18
60	47.257	53.42

## Data Availability

The data presented in this study are available on request from the corresponding author.

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
