# Peer review of "Patient Specific Numerical Modeling for Renal Blood Monitoring Using Electrical Bio-Impedance"

_sensors, 2022, doi:10.3390/s22020606_

Round 1
Reviewer 1 Report
Measurements were made at only one frequency, 100 kHz. In this range, the beta break is already showing, so maybe you should choose from the lower frequencies, or at least compare how much more error it brings into the measurements.
Measuring during a break in breathing is a serious problem for me, as it is difficult to solve when measuring next to a bed. Eliminating this would provide a very useful tool for treating physicians.
Author Response
Dear Reviewer
Thank you very much for your time and attention to assess our manuscript . We will consider all your suggestions in the next step of this study . This study suggests the real-time bed side monitoring, thus the patient will not be asked for breathing hold. However , the kidney displacement during movement can be avoided by an algorithm, which can take the signal just only on the exhaling period when the kidney is in own position, while the signal during the inhaling period will be ignored as the kidney is located far from the electode system
With respect
Dr.Mugeb
Reviewer 2 Report
The present work presents a very interesting application of bioimpedance
line 167: how many measurements were made on average? after how long?
why didn't you use an adhesive to fix the electrodes?
Obviously it must be seen as a preliminary work, in this sense a dynamic measurement could be proposed in order to overcome the movement of the kidneys during respiratory acts.
Have the tests on the subjects been carried out on different sexes, ages and possibly pathologies?
Author Response
Dear Reviewer
Thank you for your time and your attention to assess our manuscript and please find the attachment
with respect
Mugeb

Reviewer 3 Report
Hello, and thank you for the opportunity to read your manuscript describing your modeling of Electrical bio-impedence for renal blood monitoring.
You mention that the use of lower frequency can lead to seizures and polarising electronic phenomena. How many test patients was your protocol tested on? Did any of these people display any of these phenomena?
On page 2 line 63, please change kids to children
On page 5, lines 138-139 - "the second case suggests less perfusion of cortex". Suggesting? or in line with which conditions specifically?
on page 6, line 166 - "kidney replacement". Do you mean displacement? This is unclear.
Page 8, line 215 - what sort of alcohol was used? And at what concentration or v/v, w/v?
Page 9 and onwards - For clarity, please don't use US instead of ultrasound.
Page 9, line 248 - Equation 3 should be equation 4.
Page 11, line 271 - "a lot" -> suggest using numerous, or various instead.
Page 11, line 278 - "inverse problem" -> inverse function.
More generally, how many subjects have you tested this methodology on. You mention the 55kg slim young male. How old, how is slim a measurement of fat content. Have you tested this method on overweight and obese subjects? Can you provide data from another test subject that is less ideal?
Where do your measurements and calculations start to fall apart? Or, does this model perform consistently well?
Author Response
Dear Reviewer
Thank you very much for your time and attention to assess our manuscript and please see the attachment
with respect
Dr.Mugeb

Round 2
Reviewer 3 Report
Thank you for you responses, and for answering the questions that I had.
Good luck with your work - I am interested in following it going forwards.